# Reconceptualising Children's Agency as Continuum and Interdependence

**Tatek Abebe**

Department of Education and Lifelong Learning, Norwegian University of Science and Technology (NTNU), N-7491 Trondheim, Norway; tatek.abebe@ntnu.no

**Abstract:** Although the idea that children are social actors is well-recognised within childhood studies, the structural contexts shaping child agency and the everyday practices that manifest in children's social relationships with other generations are not fully elucidated. This article identifies and discusses multiple and often contradictory concepts of agency as well as a framework for re-conceptualizing it as a continuum, and as interdependent. The central argument I make is that there is a need to go beyond the recognition that children are social actors to reveal the contexts and relational processes within which their everyday agency unfolds. It is also vital to ask what kind of agency children have, how they come by and exercise it, and how their agency relates them to their families, communities, and others. The article draws on research and ongoing debates on the life worlds of children in diverse African contexts in order to critically demonstrate how their agency is intersected by experience, societal expectations, gender, geography, stage of childhood, and social maturity. In so doing, the contextualized discussions and reflections have implications to rethink childhood and child agency elsewhere.

**Keywords:** child agency; children; childhood; generation; interdependence; agency as a continuum; Africa

## 1. Introduction

The field of social studies of childhood problematizes and transforms the 'natural' category of the child into a 'social-cultural' category. It draws attention to children's social, cultural, material, and spatial worlds, and reinterprets and understands these worlds not only from the perspective of children but also from the vantage point of their everyday lives. One of the key ideas in the social studies of childhood is the recognition that children are social actors; they have agency. In fact, the concept of agency has become so pervasive that it has come to represent something that all children should have the right to exercise (Durham 2011). Children's right to exercise agency is exemplified by the application of universal, rights-based framework in the planning and provision of services targeting them (Tisdall 2015). Over the past decade, a growing body of literature within childhood studies has documented children's active contributions in the spheres of family, community, the economy, the workforce, and education (e.g., Abebe et al. 2017; Spittler and Bourdillon 2012). Studies have also drawn attention to children's engagement in popular culture, rights, activism, online participation, participation in research, and inter-generational relationships (for recent analyses on the agency of children in different realms of power and experience see (Esser et al. 2016; Oswell 2013; Spyrou 2018)). These scholars take a critical look at the notion of agency that has tended to valorise the discursive, agentful, and competent subject. As Prout (2005, p. 65) has noted, "the agency of children as actors is often glossed over, taken to be an essential, virtually unmediated characteristic of humans." The critique to move beyond binaries regarding children's agency also comes at a time when there is growing recognition of children's role in social reproduction and how their generational positions are

repositioned due to rapid social change (e.g., Abebe and Ofosu-Kusi 2016; Ansell 2016; Huijsmans 2016). Although the notion of agency is highly contested; an impasse regarding theorising the role and place of children within the agency-structure debate has recently been noted (Hanson et al. 2018). Hammersley (2016) cautions about the dangers of a simple model for agency in which children are seen to "exercise autonomous will," while paying limited attention to the complex contexts and structures that (dis-)qualify such agency. Spyrou (2018, pp. 121–22) argues that "the 'discovery' of the independent . . . child-agent has become in many ways a conceptual trap for Childhood Studies and an obstacle to its theoretical imagination." He urges scholars to "decentre" the child in order to facilitate a relational understanding of childhood across space, time, and over the life course.

In this conceptual article, I explore multiple and often contradictory ideas of children's agency. I discuss the kinds of agency children have, how they come by and exercise it, and how their agency relates them to their families, communities, and others. The central argument I make is that it is important to go beyond the recognition that children are social actors to reveal the social, cultural, material, and political contexts as well as relational processes within which their everyday agency unfolds. In so doing, I contribute to the ongoing debates about how to move forward productively in theorising child agency from relational and generational perspectives.

The article starts with an exploration of how children and childhood are given particular social and cultural meanings by drawing on research on children in diverse African contexts. The aim here is not to undertake a literature review or give 'empirical evidence' on children's lives. Instead, it is to provide examples in order to critically engage with and offer conceptual clarity regarding debates on child agency. Such exemplification are necessary not only because there are limited studies that apply 'western' theories on child agency into African settings but due to the fact that children in diverse African societies (e.g., Ethiopia, Ghana, Zambia) grow up in similar structural contexts of poverty, familial arrangements, sibling relationships, modes of socialisation, and livelihood activities. Second, I discuss the origins of, and assumptions around, the concept of children's agency, while simultaneously contesting some of the taken-for-granted assumptions around it. This is followed by a discussion of 'typologies' of agency and two interrelated perspectives in order to reconceptualize agency: agency as a continuum and agency as interdependence. The last section provides reflections on moving beyond binaries in researching agency/dependency in children's life worlds.

## 2. Unpacking 'Child' and 'Childhood'

Both 'child' and 'childhood' are value-laden concepts that vary cross-culturally and, hence, need to be unpacked. Unpacking the social and cultural meaning of childhood enables us to understand not only the context in which children are raised but also the 'values and valuations of childhood'–children's role and position in society–with implications to thinking about agency. Childhood is a phase of life that is demarcated by different life events; many societies acknowledge several developmental milestones within childhood. These milestones are sometimes celebrated formally with rites of passages during which several duties and responsibilities, symbolic and actual, are bestowed upon children. Milestones in childhood often begin with pregnancy and childbirth–a logical prelude to all further discussions about children. In Ethiopia, for example, birth and christening are important rituals in early childhood. Among Christian Ethiopians, a child is often very much longed for and considered a "gift from God" (Kassa 2017). As Poluha (2008) explains, marriage without children is thought of as troubled and partners are encouraged to divorce and remarry. At birth, the priority for a traditional birth attendant is to save the mother and then the child. A Christian woman is considered impure after birth and is purified by a religious ritual that is performed 40 days after the birth of a boy and 80 days after the birth of a girl. Christening marks the child's entry into the church, both as a member of a parish and the Christian community (Hammond 2004). When boys and girls are baptized, they are given a name that corresponds to the saint that is celebrated on the day of baptism. This name can be used in everyday life but is often replaced by another 'worldly' name that is indicative of a wish the parents have for or of the child. Naming often reflects significant events that

occurred at the time of the baby's birth. It also demonstrates familial experiences and expectations of the child's future life.

Sibling-care is a common way of socialization and 'priming' of children for collective responsibility (Nsamenang 1992). The child will be breastfed, but when the mother gives birth to another baby, the 'older' baby will be handed into the care of close relatives or older siblings until the mother's period of childbed is over. Extended family provide a protective social environment for children (Verhoef 2005; Kassa 2017). 'Voluntary' and 'purposive' fostering in which children are sent away to live with relatives as part of household socio-economic strategies is very common. Within this context, parents might be primary caregivers, but only periodically; and children develop multiple attachments with others. Children's wider relations with extended family members are both a response and adaptation to parental migration for work (Ansell and Young 2003). Few households in rural African contexts are autonomous economically or socially, and child-rearing practices are a collective venture. Households rely on an extended network of kinship that may stretch across several households and communities. These ties provide a vital mechanism for the care, training, and socialization of children (Verhoef 2005; Dyer 2007; Serpell and Adamson-Holley 2017), especially in rural areas. This co-residence and intra-familial interdependence characterize social or collective life. Children are often dutiful to family collectives; and kinship systems, not the state, dictate the social, cultural, religious and material rights of children.

These practices connected to childhood exemplify how children are defined in *relation* to members of family and community, and this cultural conceptualization of children/childhood may not resonate with the 'universal' definition of the child. In Amharic, the lingua franca in Ethiopia, many words represent the term 'child' as a stage of the life course. Yet, the term used in the translated Amharic booklet of the United Nations Convention on the Rights of the Child (UNCRC) (Save the Children Norway 2003), *hitsan*–literary meaning 'someone immature'–denotes *all* children, including infants, young children, and youth (under 18). The concept is used increasingly in public institutions like courts and schools, yet chronological age does not always match up with social and cultural understandings of childhood. First, as opposed to the UNCRC's hegemonic model of childhood, which suggests that this phase of life should be protected, cherished, and enjoyed for quite some time; the Ethiopian notion of *hitsan* indicates that this is a stage of a life course that one ought to grow out of. Anthropological studies (e.g., Poluha 2008) indicate that an Ethiopian child is no longer *hitsan* or 'immature' once he/she can distinguish between 'good' and 'bad' (6–7 years of age). Second, 'translation' of the UNCRC into local languages can result in a choice for a word that represents the subject 'child' and that, ironically, may contradict the 'progressive' notion of childhood in the UNCRC. Indeed, *hitsan* represents an 'infantalization' of diverse capacity and experiences during childhood. Equating a 'child' to *hitsan* obscures children's differentiated levels of competence, needs, and maturity in various life stages of childhood (Abebe and Tefera 2015).

Numerical age alone does not offer a full understanding of the maturity of a child; the age at which childhood ends and adulthood begins varies by culture, and over time. Yet, the definition of a child below 18 years of age implicated in UNCRC's 'global' notion of childhood demonstrates what Cook (2017, pp. 4–5) terms "the moral project of childhood," that is:

> the varied efforts over time by various parties to determine, arrange, or otherwise deem appropriate (or inappropriate) the boundaries and dimensions that make up the childhoods at hand, and thus of childhood generally. (pp. 4–5)

Notions of children's rights reflect normative and middle-class childhoods in the west and exported elsewhere through, among other things, media, colonialism, academia, international aid as well as development discourses (e.g., Millennium Development Goals (MDG 2000–2015); and Sustainable Development Goals (SDG 2016–2030)). These international policies are imbued with a particular ideology of "a once localised, western construction of the child" (Stephens 1995, p. 8), viewed as competent and capable subject (Kjørholt 2005a). Moreover, although (legal) conceptualisation of children's agency originated well before UNCRC; today's debates on children's agency and rights in

Africa narrowly centre on what children can or cannot do in the eye of law. A good example is children's involvement in paid labour, which is often restricted by law based on threshold age of entry into the labour market and whether children's participation in work interferes with their 'right to education' (Bourdillon 2017; Taye 2019). Arguably, the UNCRC participates in a moral project of childhood as it defines, promotes, and seeks to enforce generally—though often contested—shared notions of good and bad, right and wrong, and proper and improper in relation to children and childhood (Cook 2017). As the following section reveals, the UNCRC is also implicated in the glorification of neoliberal ideals around children's agency.

## 3. Conceptual Origins of Children's Agency

The concept of agency has two overlapping origins in academia and policymaking. The first origin is the 'actor-oriented approach' within social sciences (e.g., Long 2001), which suggests that human beings, including children, are neither passive recipients nor mere dependents on others or social structures, but instead social actors. Historicizing the 'agency turn' within social sciences, Asad (2000, p. 30) explains the move away, especially in the western world, from collective ideologies to a vision in which all individuals have the moral capacity and responsibility to act for themselves. The rise in the notion of agency is also linked to the spread of the belief in the autonomous and responsible subject, a key feature of contemporary neoliberal capitalism and mode of governance (Durham 2011; Hanson and Nieuwenhuys 2013; Aitken 2018). In childhood studies, the actor-oriented approach is 'translated' as children's ability to construct and determine their own social lives, the lives of those around them, and the societies in which they live (James and Prout 1997, p. 8). James and Prout (1997, p. 78) argue that through focusing on children as competent, individual social actors, we might learn more about the ways in which "society" and "social structure" shape social experiences and are themselves refashioned through the social action of members.

The second source of discussion on children's agency is the legal and moral framework presented by the UNCRC (United Nations 1989). Following the rapid ratification of the UNCRC worldwide, rights-based planning and programming for and with children placed children's rights at the top of the political agenda in local, national, and international contexts. The UNCRC views children as human beings with rights to participation, autonomy, and self-determination. For example, Article 12 gives children the right to participate in decisions that concern them. Similarly, Articles 13 and 5 focus on, respectively, the right of children to be heard, and their right to proper guidance in accordance with their 'evolving capacity.' The recognition of children's capacities to make decisions in matters that affect their lives in tandem with the principle of the 'best interests of the child' became pivotal to the acknowledgement of children's individual rights. Moreover, these rights have given impetus to a view of children as active, competent, right claiming subjects (Kjørholt 2005a, 2005b).

The above strands of ideas, that is, the academic view of children as active and knowledgeable human beings, and children as right claiming subjects in international policy, have contributed to the recognition of children's capacities and competencies. Although these two ideas originate from different sources (the former from the sociology of childhood and the latter from the International Decade for Children and NGOs movement for the ratification of the UNCRC); they became popular and entered the social sciences in early 1990. From 1990 onwards, the two perspectives coalesced and the view that children are competent social actors was consolidated.

## 4. Questioning Assumptions about Agency

There are several unexamined assumptions about children's agency. In this section, I outline three common assumptions on agency as a backdrop to the subsequent discussion that elaborates on different types and manifestations of child agency. The first assumption is linked to the role and capacity of the individual child in the society in which she or he lives, representing the agency-structure debate. There is a tendency in childhood studies to view agency as the exercise of free will against the constraints of social structures (Mizen and Ofosu-Kusi 2013; Hammersley 2016). This assumption

counterpoises the individual child to society and culture and implies that agency can be best exercised when individualism takes precedence over collective concerns. In this perspective, agency is tied to an independent selfhood, the liberation of individual as a self from cultural and social constructions and is measured against the ideal of western individualism (Durham 2011; Twum-Danso Imoh and Ansell 2015). Durham (2011, p. 152) argues that this kind of agency descends from western philosophy that privileges individual capabilities, especially the capacity of individuals to resist inequality and cultural and social expectations. It is also vested on a specific narrative of family and neoliberal ideology of personhood. Yet, as Kjørholt (2005a) cautions, the notion of childhood/children as an independent—rather than interdependent and an often-fluid socio-generational category—leads to a problematic separation of children's life from the wider social and cultural context within which it is embedded. She suggests that the popularization of children as subjects of rights on par with adults has contributed to unhelpful individualisation of children (Kjørholt 2005a). The focus on the individual also overlooks how rights and competencies, for children as well as adults, are relational and developed through participation in social practices, cultural contexts, and in social interactions (Kjørholt 2005a; Lee 2001). This is an important point because the moral framework of child rights reflects and carries several assumptions, including the premise that children are first of all individuals and only secondarily members of families and communities (Anderson 1996, p. 50). As several studies demonstrate, the desire to sustain group solidarity and interdependent life often overshadows the needs and desires of individual children or indeed any individual at all (Abebe and Tefera 2015; Kassa 2017).

Linked to the above is the assumption that child agency is universal. This assumption draws on children's 'participation rights' as imbued in UNCRC, and the recognition that they have voices that need to be heard. The assumption here is that all children can and will act in their best interest and make decisions if given the opportunity. As noted above, this is consistent with the conception of autonomous agency, which presupposes a notion of the subject as 'responsible' citizen. It also reflects liberationist notions of voice, autonomy, and the right to participation as something that all children are entitled to and capable of exercising (Spyrou 2018).

It is important to make a conceptual distinction between 'agency' and 'competence'. Competence spans a wide range of skills and attributes including physical, cognitive, emotional, social, and moral (James and James 2009). Although age is used as a proxy for assessing competence, it is often culturally relative, and children develop it through experience and exposure. Because of its association with responsibility, competence is also linked with arguments about the citizenship status of children and their rights (Lansdown 2005). In this context, children who have not reached the age of majority are not deemed to be competent to exercise the political and social responsibilities required of citizens (James and James 2009). Moreover, a view that equates children's participation rights and agency overlooks wider social, economic, and political contexts in which it unfolds (Punch 2016). Indeed, the idea that children's voices need to be elicited in, for example, research and law do not necessarily entail that they are competent and will act in their best interest in practice (Kjørholt 2005a; Alderson and Morrow 2011). Hence, any discussion of 'voice', 'agency' and 'competence' needs to be situated within the wider societal contexts that shape, enable or restricts it (Spyrou 2018). Recently it is argued that rather than striving to give children more autonomy from adults, there is a need to create an environment within which children can participate—exercise agency—together with and alongside adults (Taft 2015). Instead of focusing on the actor per se, it is useful to explore the actions and interactions of children within social contexts. This approach intersects with the recent resurgence in childhood studies to 'bring back' other generations—adults—into the debates on child agency (Hammersley 2016). We need to take a *relational* approach to children's agency, recognizing the respective roles and positions of children and adults as well as how they are connected (Wyness 2013). This point is elaborated in Section 6.

The third assumption on agency is the belief that children 'gain' agency as they mature and acquire knowledge, critical thinking, and skills through the increase of choice set before them and

through their increasing independence from their parents. Incremental agency is predicated on a unilineal notion of child development that is expected to increase as children grow in a universal trajectory from infancy through childhood into adulthood. This assumption epitomises 'valuing' agency, meaning measuring it in a certain amount or quantity (Durham 2011). It not only suggests that the behaviours and actions of children in the here-and-now are not good enough but also contributes to assessing their actions normatively, as either good or bad. Arguably, such a quantitative notion of agency needs to be problematized because agency has multiple and sometimes contradictory dimensions. As Durham (2011) notes, while agency is a recognized capacity of children, it is problematic to assess its existence (or lack thereof) in terms of 'quantity.' In other words, agency cannot be measured quantitatively; instead, it is a qualitative notion, and its manifestation can be described only contextually.

The above assumptions regarding agency reveal that it is important to go beyond the mere recognition that children have agency to instead ask *what kind* of agency they have, how they obtain and exercise it, how context shapes it, and how their agency relates to others (Durham 2011). Scholars have for long critiqued competency-based models of children's agency for expanding western, enlighment-based, neoliberal ideology of independent agency (Kjørholt 2005a; Cockburn 2013; Larkins 2014, Osewll 2013; Aitken 2018). They have also called to locate agency within the intergenerational order in ways that goes beyond the developmental, not-yet-adult, view of children's competencies. Mayall (2003) makes a useful distinction between 'actor' and 'agency.' The *actor* is someone who does something whereas the *agent* is someone who does something in relation with other people and, in doing so, makes things happen. This distinction implies that *actor* is about performativity (i.e., accomplishment) whereas *agent* is about relationality, including intergenerational relationships within which processes of social and cultural reproduction are embedded. In this sense, conceptualising children as agents means viewing them as 'doers' and 'thinkers' (Panelli et al. 2007). Thinking and doing are important components of any definition of agency, and there is much evidence that children are thinkers and doers (Ansell and Blerk 2007). As the subsequent discussion demonstrates, children exist interdependently with others. They also live their everyday life in the context of social structures, relationships, and institutions. This means that agency needs to be understood against the backdrop of wider fields of generational power. An intergenerational approach engages with how "social structure produces agency, and vice versa, acting as a bridging concept between social structure and individual action, made evident in social interaction" (Luscher 2002, p. 587). It also exemplifies what Corsaro and Eder (1990) call 'interpretative reproduction' whereby the activities children take part and social relationships they find themselves in both reproduce the social order and reshape their development and experiences in new ways.

## 5. Typologies of Child Agency

In this section, I outline and discuss the typologies of child agency in order to exemplify how different contexts shape children's agency and the ways in which children navigate these contexts. In conceptualising how economic and cultural contexts influence child agency, Klocker (2007) distinguishes between 'thick agency' and 'thin agency.' Thick agency refers to having the latitude to act within a broad range of choices and options. Thick agency can be the opportunity of girls and boys to choose the circumstances that affect their present and future lives. This includes, for example, being able to choose which school or activities to attend. This may also include the possibility to choose marriage partners. More broadly, thick agency implies choices and possibilities in the realms of material wealth, social networks, or support systems that facilitate better living conditions for children. On the other hand, 'thin agency' represents children's everyday decisions and actions that are carried out within highly restrictive contexts with few or limited opportunities (Klocker 2007). Indeed, social structures, contexts and relationships can act as 'thinners' or 'thickeners' of children's agency by constraining or expanding the range of available choices.

Yet, contexts of thin agency—poverty, shorter education, and responsibilities inside and outside the household—might coerce children to 'develop' personal agency. Studies have revealed, for example, the capacity of children who are carers of sick family members, or parents, in the context of poverty and the AIDS epidemic (e.g., Abebe 2012; Day 2017). Boys and girls who suffer poverty and parental mortality 'mature overnight' to take on responsibilities as carers and, sometimes, heads of households. Household headships and caregiving work by children in the context of AIDS subverts normative understanding of children's dependence on adults (a point I will return to below). In these settings, agency is embedded in developing resilience to cope with poverty by being involved in, for example, familial livelihood strategies. Such 'everyday agency' (Payne 2012) linked to coping is important and reflects the embeddedness of children's mundane activities and abilities to live through adversities. Although less recognized than common forms of public agency (i.e., children's discursive participation in civic institutions); everyday agency represents the daily struggle of children in the face of difficult material and social circumstances. Moreover, such constraints of childhood imply how children's 'arenas of agency' (Hutchby and Ellis 1998) are the byproduct of the socio-cultural context in which they are located. This is not only because children in such contexts are 'limited' by the generational structure and their minority status but also because they have a narrow range of identities/options to reach for as part of their everyday existence.

Another typology of agency that is exemplified by the increasing instances of child or sibling-headed households is 'ambiguous agency.' Ambiguous agency, according to Payne (2012), is the agency of children who, for example, live in child-headed households. The experiences of child-headed households are in stark contrast to established and normative conception about childhood dependency and the moral and social ideas about the kind of behaviour they should demonstrate (Bordonaro and Payne 2012, p. 366). The daily activities they engage themselves in and the spaces and places they occupy reflect what Aitken (2001) calls an 'unchild-like child' contradicting what is conventionally deemed appropriate for children. In the absence of parents or guardians, they are 'heads' of families and they need to provide for members of the household. Being children, they are excluded from any formal rights of adulthood (e.g., to marry, or receive public grants in their own right), but they carry the heavy duties and responsibilities of adulthood. Hence, the activities—agency—of children in child-headed households are 'ambiguous.' Ambiguous roles of children create limbo, an unrecognized in-between social space between childhood and adulthood. This has also implications for social intervention to support these children. As Skelton (2008, p. 166) points out, children who do not consider themselves to be children anymore and in many ways are not perceived as such by the wider society may feel that the rights defined for children do not apply to them. Similarly, children who consider themselves as 'adults' are excluded from the rights of adulthood simply because they are constricted by the legal definition of a 'child,' which often follows a threshold of biological age.

The notion of agency is often associated with African children who live in poverty and adversity (e.g., street children, child labourers, child beggars, orphans, refugees, children in households of mental illness or drug addiction). It is connected to how disadvantaged children overcome deprivation and demonstrate resourcefulness. In these contexts, agency is considered 'positive,' enabling children to strive towards self-improvement, responsibility and constructive action. Yet, agency is often contradictory. In researching the paradox of agency, Gigengack (2008, p. 205) cautions about the dangers of romanticising agency as inherently 'good.' Gigengack argues that, for example, street children may not always use their personal agency positively; they are often involved in practices that are self-destructive (e.g., theft, substance use, violence). Similarly, Hoggett remarks on the risk of valuing agency as inherently 'constructive:'

> The desire to give emphasis to the active, resilient, resourceful aspects of the welfare subject is an understandable reaction to the pathologising and problematizing of the passive and 'dependent' welfare subject . . . However, there is a danger that we slip into equating agency with constructive coping as if the two were synonymous. The point is that there is nothing necessarily constructive about agency, and we should be beware of smuggling normative

assumptions into our thinking here as if agency is good and absence of agency is bad. (Hoggett 2001, pp. 42–43)

Vulnerability is, indeed, an important reason that compels children to engage in practices that are manifestations of agency. Mizen and Ofosu-Kusi (2013) argue on the importance of considering children's perceptions of vulnerability, frailty, and need as a basis for a fuller understanding of their agency. By drawing on cases of migrant workers in Accra, Ghana, they underscore how agency is related to rejecting the normative order of the household in the face of poverty. Mistreatment by stepparents and relatives, tensions within the household, abject poverty and inability to be dependent on others—causes of child vulnerability—are some of the underlying reasons for children to leave their households/families behind and migrate for work (ibid.).

The dual dimensions of children's agency—as potential and constraint—is captured by Honwana and Boeck (2005) who argue that children in Africa are not only breakers of social order but also makers of various structures and systems of social reproduction. More often than not, children undergo, express, and provide answers to the crises of existing communitarian models, structures of authority, gerontocracy, and gender relations (Honwana and Boeck 2005, pp. 3–4). We need to move away from celebrating children's agency as a mere expression of 'resistance' or 'resourcefulness' to instead exploring the contradictory aspects and effects of agency in their lives. This is crucial because overemphasising the agency and resilience of children in overcoming adversities romanticizes poverty and individualizes that which requires collective action. It also perpetuates children's disadvantages by deflecting attention away from those with moral and legal responsibilities—government and other social institutions—to improve their life chances.

## 6. Reconceptualizing Agency

In this section, I discuss two alternative albeit interrelated approaches of reconceptualizing agency: agency as a continuum, and agency as interdependence.

### 6.1. Agency as Continuum

Robson et al. (2007, p. 135) argue that agency is an individual's capacities, competencies, and activities through which they navigate the contexts and positions of their life worlds. According to Robson et al., children exercise agency to fulfil expectations—economic, social, cultural—while simultaneously charting individual and/or collective choice and possibilities for their daily and future lives. This implies that agency is not only partial and contextual but also in flux. It is situated in practices and actions that transform both the immediate and future lives of children. This conceptualization also indicates that agency is negotiated continuously between children and families and communities as they navigate tensions between personal and collective interests.

Children's experiences of agency change depending on who they are with, what they are doing, and where they are (Robson et al. 2007). This is because their everyday lives move back and forth along a continuum of diverse experiences and changing degrees of independence-dependence, reflecting authority, rights, abilities, knowledge, responsibilities, and so on. Moreover, children may experience agency in some areas of their life but not in others (Robson et al. 2007). An example of shifting degrees of agency is captured through the roles children adopt in the context of violence and armed conflict. Child soldiers display what Honwana (2005, p. 49) calls "tactical agency" devised to "cope with immediate, concrete conditions of their lives in order to maximize the circumstances created by their military and violent environments." Tactical agency also applies to the everyday livelihood strategies of child beggars, who use creative strategies to generate income, including appearing helpless, victimized, sick, hungry, and lonely. They also sing songs and tell stories about their plight. This 'agency of victims' (Utas 2005, p. 403) reveals that when needed, children exhibit agency and turn impoverishment into opportunities. Yet, the agency of beggars in presenting themselves as 'victims' to the public during daytime is often countered by their night-time activities that involve being 'unruly' (Abebe 2009). Such changing behaviours of child beggars within the context of few alternatives for survival (i.e.,

thin agency) epitomises not only the fluidity of agency but also draws attention to the conditions of sometimes harsh socio-economic adversity (i.e., structural/systemic poverty) in which they navigate their lives.

In addition to engagement in livelihood strategies to obtain life-sustaining resources, some child beggars recruit a disabled parent or relatives (e.g., they take a blind person with mobility difficulty around in order to collect alms). This gives them relatively more *generational power* over the activity compared to an adult but a physically impaired beggar. In these contexts, children's agency depends on the interaction between personal agency—the ability to create and pursue a goal—and opportunities and constraints, as well as social *relationships* with other actors in *material* contexts. In the above example, agency is linked to the 'power' of those positioned as children to influence, organize, coordinate, and control events taking place in their everyday world (Alanen 1998). Although children face unequal adult-child power relations, some social and economic contexts may produce conditions in which this power is redefined. In other words, social and economic practices produce possibilities for a redefinition of power where children and adults can significantly influence each other.

Another example is exchange marriage, which demonstrates how agency is *negotiated* between children and adults. Abduction and/or exchange of girls for marriage is common in the socio-cultural process of reconciliation and conflict resolution in some cultures in Africa and beyond (e.g., Mebratie 2005). The practice of exchange marriage requires young boys to have sisters or female relatives to exchange. Taken at face value, exchange marriage is a male-oriented practice and girls are passive participants in the process. There appears little or no agency on the part of girls involved and collective decisions of families permit minimal individual choice. However, the situation is complex on close inspection, and there are many indirect ways in which girls may challenge the process. For example, a young girl may demand to know the person (relative) who is supposed to exchange her for a wife. This will give her the chance to negotiate and influence the choice of the family that her brother will be seeking the exchange from, and which she will eventually be marrying into. In other contexts, if the girl is not interested in her family of exchange, she may quickly marry another man while the male relative either has not yet had time to decide, or already has a wife, or is between marriages and in no hurry to consider an exchange.

However, youngsters may not always wait and follow what adults want them to do. Instead, they decide and take actions of their own even when it is against not only the will of the family but also the interests of the wider lineage as well as the normative preferences of the cultural setting. Abduction is used as a short-cut to avoid exchanges with unknown individuals. A boy may abduct a girl of his choice, and this will eventually become an issue for the family collectives of both sides to settle. However, a girl may also 'consent to abduction' by a lover to avoid being married to the person she does not know or want. In these different contexts, the agency of girls can be conceptualized in a spectrum of being 'backstage' and 'front stage.' While remaining in the background, a girl's agency is exhibited as ranging from having almost *no agency* and being a *victim* of forced abduction, to demanding to know the boy she is going to be married. On the other hand, being on the foreground, girls can also *facilitate* the abduction, thereby influencing the choice of which boy to marry as well as the process by which the marriage unfolds. In this way, children's agency can be conceptualised not only as a spectrum but also as being simultaneously located at both ends of a continuum.

Children can be simultaneously dependent and independent with respect to different aspects of their social and economic lives. For example, children might depend on parents for food, health care, schooling, and shelter (in the short-term), and to marry and establish a household of their own (in the long-term). In addition, although children have personal agency, which shapes their individual actions, this depends largely on and is regulated by familial contexts, opportunities/constraints, and interpersonal relationships. Because different forms of inter and intra-generational relationships moderate children's lives, children's needs are *interdependent* with those of their siblings, parents, and other members of their social networks. Moreover, whereas dependency on adults defines childhood;

it does not necessarily mean that adults are independent. Adults also depend on children for their social existence, care, and survival. The idea of interdependent agency is explored in-depth below.

*6.2. Interdependent Agency*

As noted above, understanding agency calls forth acknowledging the contexts and relationships within which children's activities are situated. One example that demonstrates *interdependent agency* is the way in which intergenerational relationships between adults and children play out in everyday life. For example, in rural Ethiopia, accumulation, distribution, and utilization of material resources are negotiated in a *circular interdependence* between the adults/elderly and young members of the community/family (Abebe 2008). Elderly people have wealth in the form of land and cattle, and cash. They also have authority, which is vested in their right to make policies, settle disputes, and impose sanctions. The possession of these attributes gives them the means of obligating children, but in turn, they must give up most, and ultimately all, of their wealth to their sons and daughters in the form of bride wealth, land and cattle. In this case, the agency of the elderly is interdependent and negotiated over time with those of the young. Attending to such negotiated interdependence (e.g., Punch 2015) allows us to understand material family support as the practice of agency replete with moral and social responsibility shaped in the junctures between norms, assets, and material conditions.

To substantiate interdependent agency further, I will provide examples on the everyday life of working children. Several studies of children's participation in an agency at work reveal that their contribution is valued in African communities (Spittler and Bourdillon 2012; Pankhurst et al. 2015). Nsamenang (1992) views children's active involvement in labour from an early age as a feature of West African parenting that, far from being neglectful or abusive (as it has been portrayed by the ILO), is informed by a deliberate socialization strategy of priming children for the responsibilities of adulthood. In such cultural practices, family context and gendered expectations are crucial in shaping the activities and practices of children (Spittler and Bourdillon 2012). Working children sustain households by generating income and contributing to family livelihoods. In fulfilling the needs for food and schooling, they engage in diverse economic activities: they work as daily labourers, beggars, hawkers, carers, brokers. They also assist their parents in farming and trading or allow adults to do certain activities while they attend to domestic chores. In these settings, agency is situated in daily circumstances of life, in making ends meet, or by being involved in diverse livelihood strategies for collective existence. This highlights that to single out children's agency at work overlooks the extent to which their competence is circumscribed by interdependent familial livelihoods and networks with whom mutual goals are set (Abebe 2013). Moreover, due to their significance in providing income, working children invest their economic power in ways that situate them at the centre of the household economy. They do so by contributing financially when they can, but they have to also draw on familial resources when they lack money or are unemployed. This calls for situating children's agency in context within which it unfolds: Working children's lives move back and forth along a continuum of diverse experiences in relation to changing degrees of economic independence-dependence.

Yet children's agency is an integral part of and shaped by the familial notions of care, obligations, and reciprocity. As Serpell and Adamson-Holley (2017) note, traditional African practices like sibling-to-sibling caring responsibilities are manifestations of interpersonal agency. Children's involvement in caring for, socializing, and providing informal training for younger siblings are dimensions of care that are embedded in interdependence and interactions. Because different forms of inter and intra-household relationships moderate children's lives, children in these contexts perceive their needs as interdependent with those of their siblings, parents, and other family members. While this does not contradict the principle that children should be taken seriously, it highlights the fact that their capacities are an integral part of, and are shaped by, the capability of households. Thus, to single children out as individual actors ignore the extent to which their agency—and lives—are circumscribed by intergenerational relationships as well as social and cultural contexts.

What the above examples demonstrate is that working children's agency is not an antithesis to the idea of their dependency. Nor should their ability to earn money at a young age be confused with possession of 'autonomy' or 'self-determinacy;' it is too simplistic to use the notion of (in) dependence, whether of children on adults, or adults on children. This is because, as Anderson (1996, p. 43) notes, "we are fully [independent] only when we are free from dependence on others and that freedom from dependence on others means freedom from any relations with others." Child-adult relationships should be explained in terms of *interdependencies*, which are negotiated and renegotiated in relation to the particular social and cultural context (e.g., Punch 2015). In addition, interdependent relations between children, families, and communities are dynamic and evolve with time. For instance, when working children acquire income (economic power), this tends to increase their social power, often expressed in making decisions about how to spend that money. Consequently, relationships between children, adults, and families are renegotiated accordingly, highlighting that relative dependence is not necessarily associated with a complete lack of individual agency.

Furthermore, family circumstances influence children's participation in certain activities such as education, work, or migration. In many African contexts, cultural notions of responsibility and economic circumstances affect how children make decisions of whether and how to combine work and school (e.g., Spittler and Bourdillon 2012; Pankhurst et al. 2015). When engaging in material family support practices, children in diverse contexts often describe complex layers of responsibilities for the self and immediate and extended families (e.g., Abebe 2012; Mizen and Ofosu-Kusi 2013). Families often extend support to relatives in need and sometimes even to strangers such as orphans based on a shared sense of coping with adversity. Children also talk about their creative problem-solving to manage competing responsibilities (Abebe 2008). In these contexts, a child's growth and development is not based on increasing independence and self-determination, but on increasing interdependence. Such experiences and interpretations of agency as interdependent are not uncommon in African children's life worlds and are antipode to the notion of autonomous selfhood and personal freedom.

Yet the idea of agency as interdependence is not just a critique against western, neoliberal notions of self and personhood. Rather, it is an alternative conceptual framework to theorize agency from a life course perspective. As the above examples have shown, the desire to sustain family solidarity and interdependent life often overshadows individual needs and interests, including those of children and members of their family collectives (e.g., Abebe 2008, 2013). As Punch (2015) further notes, although children in these contexts achieve relative independence at a young age, family interdependence continues over the life course. Indeed young people's agency in diverse African settings is assessed not by their ability to lead an independent life or accumulation of wealth but, instead, by their capacity to 'attract' and support dependents. If a young man or woman does not support siblings and relatives or begin to hire people and attach dependents for the house, and have children—the kind of things that begin the recognition of adulthood—that person is not seen as having agency (Durham 2011). In this sense, agency is the ability to support interdependent livelihoods and fulfil familial expectations over time. It is not just a mere manifestation of competence; it is a strategy of collective existence through which social reproduction is sustained.

## 7. Concluding Reflections

It is often taken for granted that children are social actors, that they have agency. The dominant discourse "commonly deployed within childhood studies views agency as a means to stress the capacity of children to choose to do things" (Mizen and Ofosu-Kusi 2013, p. 363). Yet, such notions of children as capable and rational actors have opened a complex field of inquiry in theorizing their life worlds. It has also produced some assumptions, which I explored in this article. Agency is not a universal experience. Instead, it is dynamic, situated, and contextual. Child agency suggests neither an innate capacity that is lost nor the rejection of the social structures that enable and constrain children's social actions.

Children's agency is often romanticized in universal, rights-based discourses and agency-centred studies in which they are recognized as competent and independent. This has led to a growth in perspectives on the social and cultural competence of children, with an emphasis on 'evidence of agency' (Durham 2011). As Durham notes, there is a need to free the concept of agency from its narrow association with free will and liberalist autonomy. Moreover, in researching childhood, the primary questions should not centre on children's competency linked to biological age or whether they can and do exercise agency. Indeed, these questions are rendered at least partially redundant because children's multiple contributions to their families and communities are so palpable. An important but under-theorized set of questions relate to the spatial, political, and material factors that shape the lives of children, the 'choices' they might confront, and the types of futures they might expect, experience, negotiate, and navigate.

This article has demonstrated that children's agency is negotiated and renegotiated with people they interact within different contexts, at different times. Children are independent and dependent at the same time, and their agency varies, depending on where they are, what they are doing, and with whom they are (Panelli et al. 2007; Esser et al. 2016). As children's lives are also intersected by such factors as maturity, gender, geography, experience, and livelihood circumstances, their agency, too, needs to be conceptualised from these vantage points as well as interdependent social relationships in which they find themselves. In other words, children agency is both constituted in social contexts and negotiated through social interaction with 'other' generations. To understand the concept of agency it is also useful to recognise that generational categories like 'child' or 'childhood' exist only in relation to—and negotiation with—other generational categories such as 'youth', 'adults' or 'the elderly.' This is not simply because children will become youth or adults and their competence is analysed relative to these stages of the life course. Instead, it is because any practice of agency takes place within the context of intergenerational relationships and the social structures that produce these relationships. Moreover, analysis of children's agency may not be limited to the analysis of childhood perse. If childhood is a relational category, other generations such as youth, adulthood, elderly have to be the object of research and theorisation as well. Childhood and 'other generations' co-determine each other, and relationships between them are not just oppositional but also productive of one another.

Perspectives that counter pose agency in dichotomous and oppositional terms (i.e., children as active, independent, competent, capable, rational versus children as passive, dependent, vulnerable, incapable, irrational) are not helpful analytically. Children's lives are better explained by and/and/and than by either/or. Children are both dependent and independent at the same time, and their agency should only be researched in the social-cultural and political-economic contexts in which they are located. The idea of interdependent agency also allows us to rethink the ways in which intervention programs for children might be designed. Individual notions of competence ignore the families in children's lives and detach already disadvantaged children from their social contexts. Children can be *empowered* if they are seen as relational beings, enabling them as well as adults they are connected with to recognize their respective role in ways that reflect their everyday life.

There is a need in childhood studies to push back against the straightforward acceptance of children as active agents, able to make decisions in their own best interests. There is also a need to move beyond popular debates about children's agency and focus on what kind of agency is deemed 'productive' for them and how the relationship they find themselves in enables or restricts it. While taking children seriously as social actors, it is important to be mindful of the ways in which "specific relations of subordination create and enable [their] capacity for action" (Hoechner 2018, p. 18). This requires acknowledging intersecting cultural, social, historical, political, institutional and material contexts of childhood. Analysis of children's agency in these multiple contexts also widens the focus. While the specific contexts within which agency unfolds in children's lived and everyday experiences remain important, this needs to be paired with an understanding of the broader social and generational structures.

**Funding:** This research received no external funding.

**Conflicts of Interest:** The author declares no conflict of interest.

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
