# Peer review of "Reconceptualising Children’s Agency as Continuum and Interdependence"

_socsci, doi:10.3390/socsci8030081_

Round 1
Reviewer 1 Report
I think the changes suggested by the reviewers have been taken onboard very well by the author with evidence of some good development of thinking and reading. The signposting in particular works to strengthen the argument and make what is being covered clearer for the reader . I had suggested in my initial feedback that the diagram of the arrow on page 11 of the article (for agency) is removed. This visual doesn't add anything, and I think should come out before it is goes any further and is published; otherwise I think it should proceed.
Author Response
Dear Reviewer
Thank you for your comments and feedback on my revised paper. I have now finalized the revision process. You have some concerns about the actual relevance of the diagram, so I have now removed it from the manuscript as your recommendation.
Reviewer 2 Report
The paper presents an interesting topic of discussion. It is given an overview and a provocative debate trying to create a new conceptualization of the child agency, against the traditional approaches.
The abstract, which contains much more than 100 words, includes the purpose, results, and global implication, it is suggested to include information about the methodology used.
The methodology used is not stated in any part of the article, so the reader is not informed how the work has been done. Only in the introduction there is a very short reference to it: “is not to undertake a literature review or give ‘empirical evidence’ on children’s lives.”
There’s a need to support the results, is there empirical evidence or literature?
As, for example, the abstract refers to a “research on the life worlds of children in diverse African contexts”, it was expected to have more detailed information that just Ethiopian reality.
The figure 1, should be improved in ordered to clearly saying what the author wrote in the manuscript: “as the diagram shows, children may experience agency in some areas of their life but not in others”, lines 389-390, but the fact is that the diagram have just the “Agency” concept, it is suggested to clearly put there the areas of children life.
Author Response
Dear Reviewer
Thank you for your comments and feedback on my revised paper. I have now finalized the revision process and hereby provide response on how I attended to your comments.
Reviewer comment | Author response |
The methodology used is not stated in any part of the article, so the reader is not informed how the work has been done. Only in the introduction there is a very short reference to it: “is not to undertake a literature review or give ‘empirical evidence’ on children’s lives.” There’s a need to support the results, is there empirical evidence or literature? As, for example, the abstract refers to a “research on the life worlds of children in diverse African contexts”, it was expected to have more detailed information that just Ethiopian reality. | I have explained how the paper draws on examples from the literature to engage with ongoing debates on child agency. The paper does not report on empirical fieldwork but instead is conceptual. It draws on nearly two decades long research, publications as well as engagement with the literature in childhood studies in diverse African contexts. The idea is to make theoretical contributions regarding debates on child agency.
|
The figure 1, should be improved in ordered to clearly saying what the author wrote in the manuscript: “as the diagram shows, children may experience agency in some areas of their life but not in others”, lines 389-390, but the fact is that the diagram have just the “Agency” concept, it is suggested to clearly put there the areas of children life.
| The diagram has been removed from the paper. The manuscript has also been revised to reflect this. |
This manuscript is a resubmission of an earlier submission. The following is a list of the peer review reports and author responses from that submission.
Round 1
Reviewer 1 Report
I suggest revising the abstract to focus more on the aims and content of the article instead of posing several questions. While the questions help to engage readers, there is less space to allow the abstract to provide an overview of the article.
Overall, I suggest revising the manuscript to use active voice.
p.1 line 22 I suggest revising to read the 'field of social studies of childhood' as this area of study does not constitute a paradigm.
p. 1 line 27 a reference should follow that sentence to support the claim that agency has come to be understood in this way. Further, by bringing in the notion of children's 'right' to exercise agency, the authors suggest agency is a right - this is confusing and detracts from what seems to be the main point of the sentence.
There is a suggestion in this paragraph (line 37) that agency is being conflated with competence. I suggest the authors clarify how they are differentiating these terms.
Section 2. The section aims to unpack 'child' and 'childhood' but quickly veers into quite detailed descriptions of rites, traditions and values related to children in a specific cultural group. This section could be much shorter and to the point with a focus on examples of how such terms are reflected (or not) across various languages and cultures.
Again in line 32, children's rights are discussed but throughout this section, I did not feel that the information being shared aligns with the section heading.
Section 3 - arguably, legal conceptualizations of children's agency originated well before the UNCRC. Again, discussions of children's rights are coagulated with the topic of agency. To suggest children have rights does not equate to a belief that all children are competent. Capacity and competency are generally viewed along a continuum under the law whereas rights are understood quite differently. I would like to see more clarity throughout this section, as well as references to support the claims being made (see for example line 173-4).
line 202 The argument being made is not logical and becomes frustrating for the reader e.g. to claim that children's voices should be elicited does not in any way suggest that they will act in their best interest. There is not presupposition of autonomous agency - merely an argument against previously held views that children are incapable of expressing their views. While I agree with the need to take a relational approach to understanding children's (and adults') agency, I find the arguments made to support this stance are overstated and lacking in clarity.
In this section, the same pattern persists e.g. the author(s) make a claim about how agency has been understood or is based on faulty assumptions but fail to support those claims. Overall, I feel this section needs to be better supported. There is a sense that the authors are trying to do too much in this one manuscript as as a result, individual arguments lack clarity and supporting evidence.
Section 5. There is a need to provide a rationale for shifting to a section on typologies of agency. The authors have to this point argued for relational understandings of children's agency. Why is there a sudden shift to introducing a typology of child agency. More signposting is necessary at a minimum, but ideally the introduction to the paper would provide a better rationale for why this information is helpful to the reader. At this point, as a reader, there is a sense that the paper is simply itemizing what is known about children's agency. A clear direction and synthesis of interdisciplinary literature is missing from this reviewer's perspective.
Section 6. Many scholars have conceptualized agency as a continuum and relational (interdependent) thus, I do find it accurate to describe this section as a reconceptualization. Further, Figure 1 does not add to the argument in any way. Why suddenly in this section, first-person language appears. I suggest revising to use active voice and first-person consistently across the paper.
Section 7 In concluding reflections, the author claims that "dominant discourse in childhood studies is that children are capable and competent". This is just not the case and once again, capability, competence, rights and agency are being conflated. Each of these should be clearly defined early in the paper and 'unpicked' to differentiate how each is used. This lack of care and clarify, reduces the readers confidence in the overall arguments being made and detracts from the potential contribution of the paper.
Reviewer 2 Report
This is in many ways an excellent paper. Its contribution to the debate around children's agency brings it up to date with very current literature, and the global dimension and thinking about relationality and inter-relations between child agents and others in their lives is a distinct and significant contribution to the field. It is well theorised and argued and well worked up. There, are, however, a few significant areas to work on:
- A core concern about this paper which needs to be developed further/explained a lot more is the way in which the African countries are used. What is the rationale for the countries chosen? The range (eg Ethiopia, Nigeria, Ghana) seem very broad. What connects these countries, for example they are not geographically close, is it the fact there are similar issues with poverty/family structure that connect them? Also, as there is no empirical research as such in this paper, the purpose and the relevance of the African contexts you provide needs to be made much clearer, in the sense that it functions to act as examples for applying the theory it seems- if this is the case this needs to be said so more explicitly. Adding a section early on in the paper which lays out a rationale for why African contexts are used, which countries within the continent are used, why they are picked, and the kinds of examples selected and why, would help decomplicate this picture and could help tighten this paper. For example, you look at an example of beggars in one local context, and birth and childhood rituals in another, and the reasoning for what is focused on in each local context needs to be made much clearer and put upfront. As it stands these examples appear to jump around a little, and there is some lack of depth to each context, touched on as it is so briefly. If the reasoning behind the use of the differing examples (begging, street children etc) is to do with thinking about agency through the lifecourse, this would be interesting and could be an aspect of the paper which is developed further and could add to the argument. Why Africa, as opposed to for example South America or Asia is used would help develop a clarity in purpose of the paper. Perhaps pointing out that the examples are, by nature particularistic would be useful, along with what such examples offer and illuminate for the purposes of this article.
- This paper set off in a way that looked as if it was going to include empirical data; some signposting and positioning that this is not the case at the start of the paper would be useful in order to justify and pave the way for the approach which is taken where individual countries and a global south/developing context is used by way of example.
-The paper needs a very thorough proofread; there are small issues with grammar, typos and subject-verb agreement in places which can get in the way of the flow of reading.
- The debates on agency and its connection to neoliberalism (if agency is seen as about the 'individual') was an interesting and innovative application of theory on agency . Perhaps making reference to Corsaro and Eder's work (2012) on the idea that children are involved in what they call 'interpretive reproduction' could be useful too, in terms of thinking about children's 'arenas of agency' (Hutchby and Moran-Ellis, 1998) in the sense that children are limited by their minority status in a majority world, and that they only have a narrow range of identities/options to reach for as part of their process of meaning making in the world. This seems particularly relevant in the socio-economic contexts you are considering here.